# Biological and Clinical Significance of Mosaicism in Human Preimplantation Embryos

**DOI:** 10.3390/jdb9020018

**Published:** 2021-05-07

**Authors:** Ioanna Bouba, Elissavet Hatzi, Paris Ladias, Prodromos Sakaloglou, Charilaos Kostoulas, Ioannis Georgiou

**Affiliations:** 1Laboratory of Medical Genetics, Faculty of Medicine, School of Health Sciences, University of Ioannina, 45110 Ioannina, Greece; ibouba@uoi.gr (I.B.); parisladias@hotmail.com (P.L.); pr.sakaloglou@gmail.com (P.S.); chkostoulas@gmail.com (C.K.); 2IVF and Genetics Unit, Dept of Obstetrics and Gynecology, University Hospital of Ioannina, 45500 Ioannina, Greece; xatzibetty@gmail.com

**Keywords:** human mosaic embryos, preimplantation genetic testing, assisted reproduction

## Abstract

Applications and indications of assisted reproduction technology are expanding, but every new approach is under scrutiny and thorough consideration. Recently, groups of assisted reproduction experts have presented data that support the clinical use of mosaic preimplantation embryos at the blastocyst stage, previously excluded from transfer. In the light of published contemporary studies, with or without clinical outcomes, there is growing evidence that mosaic embryos have the capacity for further in utero development and live birth. Our in-depth discussion will enable readers to better comprehend current developments. This expansion into the spectrum of ART practices requires further evidence and further theoretical documentation, basic research, and ethical support. Therefore, if strict criteria for selecting competent mosaic preimplantation embryos for further transfer, implantation, fetal growth, and healthy birth are applied, fewer embryos will be excluded, and more live births will be achieved. Our review aims to discuss the recent literature on the transfer of mosaic preimplantation embryos. It also highlights controversies as far as the clinical utilization of preimplantation embryos concerns. Finally, it provides the appropriate background to elucidate and highlight cellular and genetic aspects of this novel direction.

## 1. Introduction

It is widely accepted, based on single-cell studies, that the clonal cell expansion throughout development and differentiation allows individual cells to deviate—within certain limits of uncertainty—in replication proofreading, recombination, point mutation generation, methylation maintenance, histone modification, and cell cycle control [1,2]. Depending on the rate of cellular divisions and the conditions under which the cells are propagated and differentiated, errors or flaws may occur, leading to a neutral, disturbing, or harmful result [3,4]. In this review, we aim not to offer a detailed description of all possible deviations. Instead, we focus on the deviations which are relevant to the transfer of mosaic preimplantation embryos.

For the past forty years, reproductive science has managed to overcome several natural barriers and reached the current landscape in assisted reproductive technology (ART) and its clinical applications. Nevertheless, we still do not know the impact of each particular intervention on each gamete or each embryo generated by the ART process. This is due to the fact that we cannot have a spontaneous control for each ART case. Interventions, such as hyperstimulation, poor or high response, or other co-morbidities of subfertility, may inflict nuclear responses to the gametes that are extremely difficult to document or monitor [5,6]. Intracytoplasmic sperm injection (ICSI) from oligo-teratospermic men with genetic defects which affect sperm production seems to impact the incidence of congenital abnormalities and development [7]. In vitro maturation of oocytes and artificial differentiation of the sperm cells or even the use of immature gametes is also linked to adverse effects on the birth rates and defects [8]. We should have in mind that gametes do acquire nuclear changes in all possible levels of cellular organization that render them a virtually reconstructed cell machinery within a certain infinite possibility. The possible combinations of the chromosome pairs of the 23 human chromosomes (2^23^ = 8.388.604) in one parent results in more than 70 trillion possible outcomes for the offspring (2^46^). Additionally, we should not forget that this is only estimated on the basis of the homologous chromosome recombinations. Any recombination events within chromosomes or between unpaired chromosomes are not considered. The possibility of recombination outnumbers by far the number of humans on earth, which currently amounts to approximately 7,340,000,000. This is the measure of diversity with all potential recombination modes, which is a complex and diverse process implicating at least three main mechanisms: homologous, non-homologous and replicative recombination. Additionally, remarkable differences in the rate of recombinations between the two sexes and in the rate of gamete differentiation are noticed. As a matter of fact, recombinations are as high as 50% in the oocyte than in the sperm. These differences draw a striking picture of chances within the embryo and between cells [9].

Aggressive interventions, such as biopsies of the third or the fifth-day embryos, are distorting the randomness of cellular dispersion in the embryo or, most importantly, blur the distribution of mosaicism within the blastocyst [10]. Nevertheless, two of the most recent and disputed interventions, namely the spindle transfer for oocyte-inherited mitochondrial diseases as well as germline and embryo gene therapies, still have a long way to go until concordance—concerning the ethical and biological issues—is reached [11].

## 2. Chromosomal Instability and Mosaicism

In 2009, the ART scientific community was overwhelmed when Vanneste et al., 2009 in their work published in *Nature*, claimed that one of the most prominent hallmarks of tumorigenesis, namely the chromosome instability, is also a common phenomenon in the cleavage-stage human embryos derived from normal fertile women, but not in the preceding premeiotic or meiotic cell cycle stages [12]. Mantzouratou and Delhanty’s review paper [13], as well as Mertzanidou et al.’s 2013 research paper [14], demonstrated that not only chromosome instability but also extensive mosaicism are common findings in the cleavage-stage embryo. They conclude that 60 to 70% of day-3 IVF-produced embryos are mosaic with at least one segmental or whole chromosome aberration. Mosaicism is defined as the coexistence of two or more genetically different cell lines within an organism, an individual, and/or an embryo and is derived from a single zygote. This phenomenon is a prevalent characteristic of human preimplantation embryos, in which one cell lineage contains a chromosomal abnormality and the other shows normal chromosomal constitution [15]. Βased on their chromosomal profile, human preimplantation embryos can be classified into three main categories: euploids (uniformly normal complement of chromosomes), aneuploids (uniformly abnormal complement of chromosomes), and mosaics (euploid–aneuploid mosaics, aneuploid–aneuploid mosaics, and chaotic mosaic with multiple aneuploid chromosomes). The primary cause for this phenomenon is the defective cell cycle control, and the secondary cause is the parental predisposition to chromosomal instability or the genetic background. In the study by Mertzanidou et al., 2013, 71% of the blastomeres of day-3 good quality embryos were mosaic, and 29% of abnormal cells had structural abnormalities. On the other hand, when compared to the cleavage stage embryos, chromosomal mosaicism appears to be at a lesser extent in the blastocyst stage with a varying degree of mosaicism [16]. As it is well known, mosaicism is more commonly found in the placenta and is underlying the survival of severe aneuploidies such as trisomy 13 and 18 [17]. Chromosomal mosaicism is also the cause of parental disomy or isodisomy, which results from a spontaneous repair of a trisomic zygote [18,19]. Although it is diagnosed in <2% of prenatal specimens, only a small proportion of them is identified in the fetus [20].

## 3. ART and New Technologies

Embryo grading is independent of chromosomal aberrations. It is also acknowledged that a good morphological score does not rule out abnormal chromosomal content. Moreover, aneuploidies result in a marked reduction in live births and pregnancy loss and, most importantly, in vitro culture conditions and embryo manipulations may compromise spindle formation and cell division [21,22,23]. Therefore, the need for an enhanced grading system by which embryo stratification allows embryos to be transferred with the best likelihood of a positive clinical outcome, namely a healthy live birth, is now needed more than ever. The new technologies developed in genomics, metabolomics, proteomics, and time-lapse imaging could help select the best embryos. Preimplantation genetic screening for aneuploidies has been available for IVF patients for almost twenty years. It allows one to transfer embryos with the best implantation potential. Indeed, improvements in the preimplantation genetic testing for aneuploidies (PGT-A) applications, including the biopsy and subsequent analysis of multiple cells at the blastocyst stage, in addition to the introduction of high-resolution comprehensive chromosome screening (CCS) techniques—such as array comparative genomic hybridization (aCGH) and next generation sequencing (NGS)—have advanced the identification of new types of chromosomal abnormalities and different levels of mosaicism [24].

Studies have revealed that aneuploidy arises from both meiotic and mitotic errors [25]. Additionally, the increase in aneuploidy with maternal age, as already mentioned, is well documented [26]. On the other hand, mosaic embryos, are mainly generated by mitotic errors during post-zygotic cell divisions and may occur via a spectrum of mechanisms, including mitotic nondisjunction, anaphase lag, endoduplication. More importantly, mitotic errors appear to have a flat rate of incidences throughout the maternal reproductive ages [27]. The number of aneuploid cells within mosaic embryos depends on the time at which the mitotic error occurs. Embryos can be distinguished as embryos with a low or high level of mosaicism [28,29].

## 4. Embryo Biopsy and Mosaicism

Vera-Rodriguez and Rubio (2017) depicted all the possible mosaic conditions and the potential diagnosis for each condition [29]. Surprisingly, embryos diagnosed as euploid maybe mosaics were affected by segmental or whole chromosome aberrations in the inner cell mass (ICM). It is now well established that the best strategy to identify mosaic embryos is by performing a biopsy at the blastocyst stage because a smaller proportion of the total cell number is removed (approx. 5/150 = 1/30), as compared to the eight-cell stage (1/8) [30]. As the biopsy is performed on the trophectoderm (TE) only, an embryo with mosaicism in the TE and an intact ICM could be inaccurately characterized as aneuploid. A TE biopsy result may not represent the entire embryo, as for example, the unbiopsied TE cells or the ICM. Based on these assumptions, it has been postulated that an embryo diagnosed as mosaic is truly only at risk of being mosaic. In this context, according to the cell line affected (ICM or TE), blastocyst mosaicism can be subdivided into four categories: 1. Total mosaicism is observed in embryos where both the ICM and TE contain euploid and aneuploid cells; 2. ICM mosaicism with a mix of euploid/aneuploid cells found exclusively in the ICM; 3. TE mosaicism with a mosaic population of euploid/aneuploid cells confined exclusively in the TE, and 4. ICM/TE mosaicism in embryos where all cells of the ICM are aneuploid while the TE cells are euploid and the other way round (ICM euploid and TE aneuploid).

## 5. Implantation Potential of Mosaic Embryos

Embryos diagnosed as mosaic have the potential to implant and develop into healthy babies [31,32,33,34,35]. The mosaic blastocysts transfer results can be categorized depending on the embryo transfer outcome. Munné et al., 2017, reported that, with the advent of NGS, 41% of the mosaic embryos resulted in ongoing implantation [33]. Complex mosaic blastocysts and embryos with >40% abnormal cells had a lower ongoing implantation rate (IR) than other mosaics. As far as ongoing pregnancy rates (OPRs) are concerned, Kushnir et al., 2018, reported higher OPR (63.3% vs. 39.2%) and lower miscarriage rates (MR) (10.2% vs. 24.3%) with euploid embryo transfers when compared to mosaic embryo transfers also utilizing NGS [36]. Spinella et al., 2018, who used either NGS or aCGH techniques, observed that embryos with lower aneuploidy percentage (<50%) have a similar clinical outcome compared to euploid blastocysts [34]. In this study, mosaic embryos with higher aneuploidy (>50%) showed a significantly lower IR (24.4% vs. 54.6%) as well as lower clinical PR (15.2% vs. 46.6%). Similarly, a retrospective cohort study reports the pregnancy outcomes after transferring mosaic or euploid embryos using aCGH [37]. Although aCGH has already been in practice for the past ten years and despite the fact that it is not a deep sequencing approach, as NGS is, the results of this technique are promising and worth mentioning. The transfer of 102 mosaic embryos resulted in 46.6% live births, contrary to the transfer of 268 euploid embryos which resulted in 59.1% live births, respectively. Furthermore, the same cohort of segmental only or whole chromosome mosaic embryo transfers were compared to euploid embryo transfers. The results of this subgroup analysis favored the segmental mosaic transfers (48.3% versus 43.5% (*p* < 0.026)). Similarly, in a recent multicenter prospective study, the authors demonstrated that mosaic embryo transfers compared to euploid embryos and non-PGT transfers revealed a significantly lower clinical PR (40.1% vs. 59.0% vs. 48.4%) and higher MR (33.3% vs. 20.5% vs. 27.4%), respectively [38].

The question that needs to be addressed relates to how the mosaic embryo transfer is compatible with a positive pregnancy outcome and the delivery of a healthy baby. Several explanations have been proposed for this outcome. Firstly, the self-correction hypothesis, by which a mechanism downstream the blastocyst stage, has been proposed [39]. In their study, they constructed a mouse model to determine chromosome-mosaicism and mosaic embryos’ developmental potential. They observed that aneuploid cells—when compared to euploid ones—had reduced proliferation levels and an increased apoptosis rate. During development, these aneuploid cells will be lost, and the number of euploid cells will increase, leading to a complete rescue of the embryos. They also demonstrated that aneuploid cells in the TE and the ICM have different behavior. They found that—contrary to euploid cells—aneuploid cells in the ICM are eliminated by apoptosis, while aneuploid cells in ΤΕ are eliminated due to reduced proliferation. This study also showed that mosaic embryos with low levels of aneuploid cells—contrary to mosaic embryos with a high mosaic level—had an increased chance of being developed into healthy babies. The different behavior of aneuploid and euploid cells was also demonstrated by Victor et al., 2019 [35]. The subsequent conclusion was that cell proliferation and death levels are considerably higher in mosaic and aneuploid blastocysts than euploid blastocysts.

Popovic et al., 2020, reviewed studies that showed that TE biopsies are not always concordant with the chromosomal constitution of the ICM in mosaic embryos [40]. It is evident that the percentage of mosaicism in the TE cells biopsied cannot be extrapolated to the whole embryo. An embryo diagnosed as mosaic is truly only at risk of being mosaic. Popovich reported concordance between the TE and the ICM in 62.1% of the embryos analyzed. Gleicher et al., 2017, used a mathematical model and calculated that the biopsy of 27 cells from the embryo allows a true representation of the entire embryo [41]. This is made under the assumption of the even distribution of aneuploid cells throughout an embryo and without considering the clonal distribution. Therefore, a biopsy of five to ten cells cannot accurately determine the embryo’s ploidy status for clinical use. Thus, the percentage of mosaicism reported depends on the distribution and ratio of aneuploid cells, and the biopsy outcome is only relevant to the biopsy itself. Thirdly, clinical treatment protocols, laboratory handling, or technical aspects in embryo culture may underly the high frequency of mosaic embryos that is not entirely relevant to spontaneous embryonic development. Furthermore, unequal threshold, cut-offs, different platforms, the lack of a standardized practice to interpret or report PGT-A results, artificial noise, an artifact in whole genome amplification or sequencing reactions, or even suboptimal biopsy collection could be the reasons for the varying percentages of mosaicism observed between the study groups [42]. Another aspect of the discordance between the TE and the ICM could be the synchronization of the TE cell division with the ICM cell division, which may also make the biopsied TE less representative than the ICM integrity.

Based on mouse model findings [39], it has been shown that aneuploid embryos may self-correct downstream by extruding abnormal blastomeres as cell debris [43]. Recently, in another research setting, where single-cell genomic data (scRNA-seq) were used to quantify mosaicism in human preimplantation-stage embryos, it was concluded that low-level mosaicism is a frequent phenomenon, whereas high-level mosaicism is relatively uncommon [44]. Single-cell analysis of human embryos reveals diverse patterns of aneuploidy and mosaicism. These observations demonstrate that mosaicism in blastocyst-stage embryos is a common characteristic that can be found in almost all embryos. It seems that early-stage embryos are dynamic systems with the ability to self-correct.

## 6. Mosaic Embryos and Pregnancy Outcome

ESHRE launched a questionnaire (20 February 2020) and addressed it to its members as a first step to collect and process opinions, views, and attitudes towards a final resolution or recommendation regarding the transfer of mosaic embryos. Nevertheless, a detailed presentation of the literature is crucial to the ART scientific community in order not only to shape a view but also to formulate a sound informed consent with legal and moral value for the patients. More extensive and larger studies must examine cumulative pregnancy outcomes to provide an exact representation of the actual effectiveness of PGT-A. Last year, Munné et al., 2020, published a retrospective study in which they compared the clinical pregnancy outcomes of blastocyst transfer after applying PGT-A via aCGH or NGS techniques to almost 3000 PGT-A cycles [45]. This study was expected to shed more light on the issue of mosaic transfer and provide firm conclusions. They concluded that, although NGS-classified euploid embryos have a higher ongoing implantation rate, the OPR per cycle were similar to the rates of aCGH. If NGS-classified mosaic embryos reached term, they were found to be euploid in cases where karyotype analysis was available. Moreover, embryos that carry uniform aneuploidies affect the entire chromosome and could not implant. The implanted embryo ended up in a chromosomally abnormal live birth.

In the first virtual meeting of ESHRE, Spinella et al., 2020, in their multicenter study, reported the clinical outcomes of 822 mosaic embryos, which were transferred at the blastocyst stage [46]. These outcomes enhanced previously published results [34]. Embryos were classified as mosaic when abnormal cells were identified within the 20–80% range of aneuploidy in the TE biopsy. This extensive analysis showed that embryos with different chromosomal mosaicism patterns presented a distinct set of clinical outcomes. They also demonstrated that mosaic embryos’ reproductive potential is affected by the complexity and the number of euploid cells in the TE biopsy. Compared to aneuploid mosaic embryos with one or two affected chromosomes, the embryos with a mosaic segmental aneuploidy had the best outcomes (implantation *p* < 0.0001, OPR/BR *p* < 0.0001). However, the implantation and the OPR/BR were less favorable compared to the euploid control group (51.3% vs. 61.1%, *p* = 0.0004 and 42.6% vs. 52.7%, *p* = 0.0003, respectively). The group with complex mosaic aneuploidy had the least favorable outcomes. Given these results, it is possible to draw a better stratification of mosaic embryos.

These findings were consistent with the results previously obtained from the transfer of 100 mosaic embryos carried out in a single-center study [35]. This prospective study concluded that embryos carrying a single segmental abnormality should be preferred, followed by those with less severe mosaicism (45% vs. 36.4% IR and 39.4% vs. 27.3% fetal heartbeat, respectively). Most recently, Viotti et al., 2021, presented their analysis which was based on multicenter data of 1000 transferred mosaic embryo outcomes [47]. Thus far, this is the largest dataset with mosaic embryo transfer outcomes. They confirmed that combined mosaic embryos (segmental and chromosomal abnormalities) have statistically significant lower implantation (45.5%) and higher OPR/BR (37%) than euploid embryos (57.2% and 52.3%, respectively). They also found that the level and type of mosaicism significantly affects the embryo transfer outcome. Mosaic embryos with segmental aneuploidy had significantly better clinical results, followed by the group with one affected chromosome and by the group with three or more affected chromosomes (complex group) (implantation 51.6% vs. 46.4 vs. 30.4% and OPR/BR 43.1% vs. 34.8 vs. 20.8%). Furthermore, no significant differences between mosaic monosomies and trisomies were observed regarding the ploidy status of the mosaic embryos. The authors concluded that mosaic embryos also develop into physiologically healthy babies and proposed a classification system for these embryos. Finally, this ranking system can be accessed as a freely available web-based tool (https://embryo-score.web.app accessed on 27 April 2021).

According to the recently published review [48], more than 100 live births have been reported after mosaic embryo transfer. There were no detected differences regarding the birth weight, preterm delivery rate, or risk of congenital malformations in the examined newborns. However, much more data concerning perinatal and long-term neonatal outcomes born from mosaic embryos are imperative to draw definite conclusions and to provide optimal clinical guidance.

## 7. Criteria for Mosaic Embryo Transfer

PGT-A has commonly been suggested for couples with an increased risk of embryonic aneuploidy. Hence, the most frequent indications include advanced maternal age (often defined as 35 years), repeated implantation failure, recurrent pregnancy loss, and severe male factor infertility. However, these indications remain controversial since there is insufficient evidence in the scientific community. Studies are mostly limited to randomized controlled trials, and they include good prognosis patients with multiple blastocysts available. The NGS-based Single Embryo Transfer of Euploid Embryo (STAR) trial showed that PGT-A did not significantly improve the PR per embryo transfer in women aged 25–40 years [49]. However, this multicenter randomized controlled trial (RCT) demonstrated a significant increase in OPR in a subgroup of women aged 35–40 years. Although these results did not reach the expected superior implantation rate, the putative loss of competent embryos could be attributed to the biopsy technique and the false positive or false negative misclassification of the blastocysts [50].

While it is well accepted that mosaic embryos have different outcomes compared to euploids, there is still no consensus on the mosaic characteristics that could affect the pregnancy rate. However, as already mentioned, data suggest that the percentage of abnormal cells detected in the TE biopsies is the principal indicator of viability. Embryos with a low proportion of abnormal cells result in viable, chromosomally normal ongoing pregnancies, while high-level mosaics result in fewer viable pregnancies [34]. Specific chromosomes implicated and the number of affected chromosomes in the biopsied cells have been shown to have a significant impact on the clinical outcomes [51]. Today, the available data on pregnancy outcomes are limited. Doubts remain as to which infertility group and which characteristics of mosaicism (numerical, structural, or both) correlate with the clinical outcomes. Moreover, the potential risks which are associated with mosaic embryo transfer are still unknown. Should embryos classified as mosaic be transferred? Recently, the transfer of an embryo with 35% mosaicism of monosomy 2 resulted in a mosaic offspring’s life birth, showing 2% mosaicism of monosomy [52]. This highlights the importance of counseling couples appropriately, especially those who have only mosaic embryos.

In this context, the Preimplantation Genetic Society (PGDIS) released an updated position statement on the transfer of mosaic embryos [53]. They addressed the consideration to transfer a mosaic blastocyst and discussed two options: (1) initiate a further PGT-A cycle to increase chances for an euploid embryo transfer and (2) proceed to the transfer of blastocysts with the lower level mosaicism after appropriate counseling. Briefly, laboratories recommended to utilize validated NGS platforms. This new application can accurately and reproducibly identify as low as 20% mosaicism in a sample. The suggested lower cut-off point for mosaicism classification should be considered a continuous risk gradient, ranging from lower risk at 20% to higher risk towards 80%. This means that PGT-A samples with <20% aneuploidy could be classified as euploid, whereas samples with >80% aneuploidy as aneuploid. For mosaic embryos within the 20–80% range of aneuploidy in the TE biopsy, a transfer could be considered. However, it should be mentioned that the value of 20% represents the sensitivity threshold of NGS platforms to detect other lineages in a TE biopsy sample. Moreover, the PGDIS denotes that these transfers should be carried out with caution. It is clearly stated that euploid embryos should always be prioritized for transfer over those with a mosaic result. When euploid embryos are not available, the transfer of mosaic embryos should be carried out according to the suggested PGDIS grading system. This risk assessment system is based on the specific aneuploidies reported in embryos. It refers to the level of mosaicism, the chromosomes involved (such as 13, 14, 15, 16, 18, 21, 22, X, Y), the aneuploidy status, and the presence of complete or segmental chromosomal abnormalities.

Munné et al., 2020, recommended a mosaic embryos’ classification into a high-level and a low-level group and prioritized single segmental mosaics for transfer over other mosaic types [45]. Viotti et al., 2021, also identified a ranking system for mosaic embryos in the clinic [47]. An alternative set of recommendations has been proposed by Grati et al., 2018 [51]. They established a practical scoring system based on mosaic patterns observed in prenatal chorionic villus sampling and conception products. However, it is still not clear which infertility groups might benefit from PGT-A.

While new technological innovations brought important improvements in reliably detecting various types of genetic errors, such as mosaicism, the interpretation of PGT-A results faces new challenges. The selection of embryos for transfer is more complex, and the need for defining more specific criteria for a clinical diagnosis is more evident. It is important to examine and address the limitations of this new technology and set new guidelines in applying uniform reporting practices. It is necessary to recognize that there is great variability regarding the definitions and transfer thresholds of mosaic embryos and that the biopsied samples may not always represent the chromosomal state of the entire embryo. Furthermore, damage or loss of blastocysts from the TE biopsy and errors occurring during the genetic analysis of the small amount of DNA may impact the reliability of mosaic diagnosis. Due to these technical and inherent limitations, some normal embryos with the potential for normal euploid pregnancies, if transferred, are discarded after PGT-A. These false-positive results can cause a decrease in life birth rates.

The introduction of NGS technology allows for the identification and reporting of intermediate chromosomal copy numbers. Therefore, there is a trend towards changing terminology on the definition of mosaicism. The American Society for Reproductive Medicine (ASRM) committee uses the phrase “embryos with intermediate copy numbers” when they refer to embryos diagnosed as mosaics [54]. A published opinion by Paulson and Treff, 2020, proposed that the latter term is more accurate [55]. The designation “mosaic” should be replaced since it is inaccurate and misleading about its clinical significance.

As mentioned earlier [48], more than 100 healthy life births after mosaic embryo transfer have been reported. Based on these observations, it can be concluded that embryonic trophectoderm mosaicism may represent a normal variant of early embryo development and not a pathological feature. On the contrary, one cannot oversee that low-level mosaic embryos appear to have better clinical outcomes than mosaic embryos with high-level mosaicism. Besides, embryos categorized as low-level mosaics have significantly poorer clinical outcomes than the euploid group.

It is important to point out that there are limited data regarding neonatal and postnatal outcomes of mosaic embryo transfers. The results in regard to IR, OPR, and LBR are heterogeneous. This highlights the need for consistent follow-up data after transferring mosaic embryos with clinical and genetic outcomes. The conduction of larger-scale multicenter follow-up studies will contribute to the risk assessment of mosaic embryo transfer and help find a balance between increasing the rates of favorable clinical outcomes and decreasing the exclusion of embryos with implantation potential.

However, a recently updated Practice Committee Opinion of the ASRM and the Society’s Genetic Counseling Professional Group (GCPG) does ‘not endorse, nor does it suggest that PGT-A is appropriate in all cases of in vitro fertilization’ [54]. The importance of patient counseling must be emphasized, and flexibility and individualized treatment strategies should be strongly considered in clinical practice combined with a polyparametric approach to reach decision making.

## 8. Mosaicism and Segmental Aberrations

An interesting insight into the type of chromosomal abnormalities and mosaicism was presented by Fiorentino et al., 2020 [56]. They demonstrated the results of a large multicenter study, in which they examined the pattern and prevalence of chromosomal constitution in 2280 mosaic TE biopsied embryos. In concordance with other studies, they observed that 25% of mosaicism affects segmental gain or loss [33,57]. Additionally, the chromosomes involved in mosaic aneuploidy were different from those involved in segmental mosaicism. However, due to each segmental abnormality’s rarity, conflicting data exist on segmental mosaicism’s clinical impact [33,35]. Therefore, there are no specific guidelines in regard to segmental mosaicism. According to the recently published guidelines, the clinical implication of transferring embryos with mosaicism and/or de novo segmental abnormalities (uniform or mosaic) is not entirely understood [58]. It is stated that the transfer of embryos with these abnormalities could probably result in first-trimester miscarriage or an unbalanced life birth. More studies regarding segmental abnormalities’ etiology will be necessary to provide the best decision-making process concerning embryos with such anomalies.

A quantitative and qualitative analysis of segmental aneuploidies in the TE biopsied samples was performed by Insua and colleagues [59]. They concluded that pure segmental aneuploidy is chromosome-dependent with an apparent topographic effect. It is also independent of maternal age, and it is not related to clinical or embryological parameters, but it shows an association with blastocyst morphology. Furthermore, a low concordance of segmental aneuploidy between the TE biopsy and the ICM has been observed [35,60]. These findings highlight the differences between the respective molecular mechanisms which lead to segmental aneuploidies and the mechanisms which are associated with whole chromosome aneuploidies.

Moreover, most segmental aneuploidies have a mitotic origin and appear during the first few mitoses following fertilization. Only one-third of segmental errors are of meiotic origin [61]. In the same study, it was also concluded that specific genetic loci present a higher chance of segmental abnormality due to heterochromatin composition in these regions. The sites of chromosome breakage do not appear randomly, but they tend to occur at distinct loci. They seem to originate from faulty DNA double-stranded breaks due to endogenous and/or exogenous factors.

Another interesting aspect of the paternal impact on segmental mosaicism is that severe oligozoospermia shows the highest incidence in preimplantation embryos. Segmental aneuploidy indicates that segmental gains and losses are mostly paternally derived [62,63]. According to Coll’s results, paternal age appears to be the only factor that significantly and independently increased mosaicism incidence [57]. All these observations can give insight into the understanding of chromosomal mosaicism’s origin and nature. They also shed light on the type of mosaicism that should be expected according to the infertility type and the parental factor involved.

## 9. New Technologies and Non-Invasive PGT-A

Depending on the new comprehensive aneuploidy screening diagnostic methodologies, the technical processing, and the embryonic stage, mosaicism’s incidence varies widely between studies. Initial investigations and diagnostic applications for PGT-A were based on aCGH technology and variable resolutions depending on the number of probes and chromosome coverage. More recently, NGS has revolutionized our view on single cells and has given a deep insight into neighboring cells in the same tissue. Deep sequencing is an approach to discriminate between individual cells or even concordant twins as distinct individual humans. Although several data demonstrated similar implantation and pregnancy rates using aCGH or NGS, recent reports support that NGS technology can identify embryos with chromosomal mosaicism and segmental aneuploidy more accurately than aCGH [64].

The application of new technologies with higher sensitivity and dynamic reading range gives a unique insight into the chromosomal abnormalities’ spectrum. Due to the evidence of potential damage caused by current biopsy techniques and the non-concordance of the TE and the ICM, PGT-A will always have a level of clinical misdiagnosis and uncertainty, even in the presence of the most accurate genetic analysis. To avoid these limitations, attempts have been made to make PGT-A less invasive and develop sophisticated algorithms to implement new technologies. DNA in blastocoel fluid (BF) and spent culture medium (SCM), mitochondrial DNA, metabolomics, assessment of nutrient consumption (glucose), proteomics, and lifetime imaging microscopy are promising approaches. They could be used as a predictive biomarker of euploid embryo competence. Indeed, multiple layers of different technologies could come into one platform, and dedicated algorithms might produce a more global insight into the embryo. Three main non-invasive DNA sampling methods have been tested and developed: 1. Analysis of BF alone, which has been labeled minimally invasive, as a procedure is still required; 2. Analysis of media alone, which is genuinely non-invasive, and 3. Combined analysis BF + SCM, which is also minimally invasive.

Today, several studies regarding the efficiency of non-invasive PGT (niPGT) protocols have been promising and sometimes contradictory (reviewed by Leaver and Wells) [65]. The presence of small amounts of DNA in the BF amplified by PCR was first detected by Palini et al., 2013 [66]. This source of embryonic DNA in blastocoele fluid is usually removed before embryo vitrification to protect the blastocyst from membrane-damaging ice crystal formation and improve embryo survival rate after vitrification [67]. It has been proposed that the BF DNA originates from cell death by the apoptosis of trophoblastic cells or the ICM of the developing blastocyst. An interesting observation of Magli et al., 2019, was that BF of aneuploidies were more likely to amplify, something that possibly reveals more DNA from aneuploid cells [68]. This could mean that the amount of cfDNA in BF might be predictive of the embryo’s ploidy status. Nevertheless, although BF-DNA can be successfully amplified and subjected to NGS, there is increased discordance between the ICM and the TE. Therefore, blastocoentesis requires improvement in sampling, processing, laboratory protocols and aneuploidy calling algorithms. Furthermore, the BF analysis is obtained by aspiration with a thin micropipette and, therefore, it is not considered a true non-invasive approach.

Another DNA source for the non-invasive genetic evaluation of preimplantation embryos involves the analysis of SCM, which is based on sequencing DNA released into the culture medium from the TE and the ICM during the latest stages of preimplantation development [69]. This assessment of ploidy status is considered to be a genuinely non-invasive approach. Hence, DNA in the SCM will enable the sampling of both the ICM and the TE, whereas the TE biopsy’s DNA will only represent the TE’s ploidy status. Rubio and colleagues described the most comprehensive study to date for truly niPGT [70]. In this multicenter prospective study, the analysis included 1301 blastocysts, and the concordance rate between embryonic cell-free DNA and corresponding TE biopsies reached almost 80%.

Recently published studies hypothesized whether the combination of BF and SCM from a single embryo could improve the PGT-A efficiency [71,72]. The study by Kuznyetso et al., 2018, revealed that combining these two DNA sources increases the quantity and quality of the total cfDNA amount in the final sample, which further shows high amplification and concordance rates [71]. The amplification of DNA was 100% successful, and the ploidy and whole chromosome copy number concordance rate reached the level of 100% and 98.2%, respectively. It is noteworthy that this minimally invasive approach allows the BF to be expelled from the blastocoel cavity without the additional step of BF aspiration with an ICSI pipette.

Due to the inconsistent results, more clarity is required regarding the mechanisms that release embryonic DNA into the surrounding fluid. It must be clarified if the SCM and/or BF accurately represent the blastocyst’s actual ploidy status. In addition to biological issues that need to be answered, technical complications can affect the accuracy of this approach [73]. These include the DNA’s potential degraded nature, which results in a variable amount of heterogeneous-size cell-free DNA. The quantity and quality of available starting material may compromise the successful amplification and even lead to allele drop-out. Notably, DNA in the SCM appears to be of superior integrity and in greater quantity than the DNA detected in the BF. Another important consideration is the occurrence of mosaicism and the detection of DNA from granulosa and cumulus cells or polar bodies in the SCM. Moreover, controversial results between study groups may relate to differences in technical parameters such as processing, analysis, and data reporting. However, niPGT-A may represent a large part of the future of IVF diagnosis and treatment. It seems that technological developments and bioinformatical approaches could overcome these limitations and help niPGT-A reach a similar amplification level and informativity as conventional PGT-A. However, the challenges in mosaic aneuploidy calling remain.

The technological development and advances in genetic evaluation and the increasing availability of big data result in a more complicated interpretation and classification of the human blastocyst. The use of artificial intelligence (AI) through machine learning is being intensively researched. Recently, AI was applied to big PGT data sets, which could help detect embryo mosaicism with higher efficiency and accuracy. AI in PGT has been reported to improve pregnancy outcomes using a second-generation AI platform [74]. These sophisticated bioinformatic approaches aim to eliminate subjectivity and variability by utilizing mathematics and statistics and to provide an algorithm-based embryo selection and assessing system. As the science surrounding several areas of reproductive medicine (sperm and oocyte selection, morphogenetics, genomics, transcriptomics, proteomics, prediction of live birth, stimulation protocols) continues to improve, a range of these approaches will be combined in order to optimize the efficiency of IVF in the future.

## 10. Conclusions and Future Perspectives

In the light of recent reports of mosaic transfers with or without clinical outcomes, the expansion of clinical indications to consider mosaic embryos as potential candidates for clinical use is striking. Reactions to a future formal resolution permitting mosaics to be scrutinized for transfer introduces new rules of engagement in this open discussion. Before we orientate ourselves as pros and cons, we should examine the whole spectrum of implications and approaches to make mosaic transfer safe and sound. Therefore, debate is still ongoing. The identification and designation of the subgroups of mosaic that are really viable and worth transferring are of extreme importance. Given the current state of observations, genetic technology is gradually embracing NGS for most diagnostic applications. In addition, NGS can access free DNA molecules and quantify with deep reading whatever is included in the sequence library.

Hybrid approaches involving biopsied blastocyst cells, free blastocoele DNA, spent media free DNA, and time-lapse videography throughout preimplantation development may help to elucidate and distinguish the clinical groups that are mild mosaics. Their aberrations do not affect the ICM of the blastocyst and have the optimal pregnancy outcome.

Our view is that just one sampling material is not enough to segregate euploid from aneuploid in the few cells that give rise to the ICM and the resulting embryonic disk. A more detailed algorithm based on cellular and free DNA coupled with NGS analysis would probably provide extensive details on the percentage, topography, and the type of mosaicism.

## Data Availability

Not applicable.

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
