# Peer review of "Biological and Clinical Significance of Mosaicism in Human Preimplantation Embryos"

_jdb, 2021, doi:10.3390/jdb9020018_

Round 1

Reviewer 1 Report

The authors  in this review article discuss  the biological and clinical significance of mosaicism in IVF

  • (17) Abstract. “ …..there is growing evidence that are viable, and have capacity…….” (I would delete viable)
  • Please define the definition of mosaicism   and  explain  kinds of mosaicism in embryos  for clinicians . Discuss  the definition of mosaicism according to ASRM and especially  PGDIS criteria. Please explain critically PGDIS criteria  for “so called”  mosaic, euploid and aneuploid embryos.
  • 221-223 – Please add more clinical data form these studies including implantation rate, pregnancy rate etc. (Spinella et all, an Viotti et al)

    It should be mentioned that mosaicism could be normal and physiological phenomena in human embryos and probably all or majority of embryos are mosaic ?? Do we PGTA laboratories  overdiagnose or misdiagnose  something?  Do  practitioners  “lost babies ” discarding  an embryos from FET ? It would be interesting  to see  more disscuscion and authors opinion in this review.  
    All the best  PR. 

Author Response

Response to Reviewer 1 Comments

 The authors in this review article discuss the biological and clinical significance of mosaicism in IVF 

Point 1

(17) Abstract. “ …..there is growing evidence that are viable, and have capacity…….” (I would delete viable)

Response 1

Thank you for pointing this out. We agree and have deleted ‘’are viable and’’ from the Abstract, Line 17.

The new sentence reads as follows:

In the light of published contemporary studies, with or without clinical outcomes, there is growing evidence that mosaic embryos have the capacity for further in-utero development and live-birth”.

Lines 15-17, page 1.

Point 2

Please define the definition of mosaicism   and  explain  kinds of mosaicism in embryos  for clinicians . Discuss  the definition of mosaicism according to ASRM and especially  PGDIS criteria. Please explain critically PGDIS criteria  for “so called”  mosaic, euploid and aneuploid embryos.

Response 2

We agree with the reviewers’ comment to define the term “mosaicism”. We have clarified the definition in lines 82-91, page 2 as follows:

“Mosaicism is defined as the coexistence of two or more genetically different cell lines within an organism, an individual, and/or an embryo and is derived from a single zygote. This phenomenon is a prevalent characteristic of human preimplantation embryos, in which one cell lineage contains a chromosomal abnormality and the other shows normal chromosomal constitution [15]. Βased on their chromosomal profile, human preimplantation embryos can be classified into three main categories: euploids (uniformly normal complement of chromosomes), aneuploids (uniformly abnormal complement of chromosomes), and mosaics (euploid-aneuploid mosaics, aneuploid-aneuploid mosaics, and chaotic mosaic with multiple aneuploid chromosomes).”

 In addition we have added the respective reference:

  1. Taylor, T.H.; Gitlin, S.A.; Patrick, J.L.; Crain, J.L.; Wilson, J.M.; Griffin, D.K. The Origin, Mechanisms, Incidence and Clinical Consequences of Chromosomal Mosaicism in Humans. Hum Reprod Update 2014, 20, 571–581, doi:10.1093/humupd/dmu016

 Furthermore we agree with the reviewers’ recommendation to

  • discuss the definition of “mosaicism” according to ASRM and especially the PGDIS criteria
  • critically explain the PGDIS criteria for the “so called” mosaic, euploid and aneuploid embryos.

 We added the following paragraph:

Lines 322-339, page 7.

Briefly, laboratories, recommended to utilize validated NGS platforms. This new application can accurately and reproducibly identify as low as 20% mosaicism in a sample. The suggested lower cut-off point for mosaicism classification should be considered a continuous risk gradient, ranging from lower risk at 20% to higher risk towards 80%. This means that PGT-A samples with <20% aneuploidy could be classified as euploid, whereas samples with >80% aneuploidy as aneuploid. For mosaic embryos within the 20–80% range of aneuploidy in the TE biopsy, a transfer could be considered. However, it should be mentioned that the value of 20% represents the sensitivity threshold of NGS platforms to detect other lineages in a TE biopsy sample. Moreover, PGDIS denotes that these transfers should be carried out with caution. It is clearly stated that euploid embryos should always be prioritized for transfer over those with a mosaic result. When euploid embryos are not available, the transfer of mosaic embryos should be carried out according to the suggested PGDIS grading system. This risk assessment system is based on the specific aneuploidies reported in embryos. It refers to the level of mosaicism, the chromosomes involved (such as 13, 14, 15, 16, 18, 21, 22, X, Y), the aneuploidy status, and the presence of complete or segmental chromosomal abnormalities.

Point 3

221-223 – Please add more clinical data form these studies including implantation rate, pregnancy rate etc. (Spinella et all, an Viotti et al)

Response 3

We agree with the reviewers’ comment that more clinical data from these studies should be included (implantation rate, pregnancy rate) and appreciate the encouraging suggestion.

Spinella’s latest data were presented at the ESHRE meeting in 2020 and Viotti et al 2021 just published the results they presented at the ASRM meeting in 2020.

According to your suggestion, we added additional clinical data as follows:

Lines 160-164, page 4.

“Spinella et al., 2018, who used either NGS or aCGH techniques, observed that embryos with lower aneuploidy percentage (<50%) have a similar clinical outcome compared to euploid blastocysts [34]. In this study, mosaic embryos with higher aneuploidy (>50%) showed a significantly lower IR (24,4% vs 54,6%) as well as lower clinical PR (15.2% vs. 46.6%).”

Lines 246 – 259, page 6

We incorporated the bold highlighted phrases:

“In the first virtual meeting of ESHRE, Spinella et al., 2020, in their multicenter study, reported the clinical outcomes of 822 mosaic embryos, which were transferred at the blastocyst stage [42]. These outcomes enhanced previously published results [33]. Embryos were classified as mosaic when abnormal cells were identified within the 20–80% range of aneuploidy in the TE biopsy. This extensive analysis showed that embryos with a different chromosomal mosaicism pattern presented a distinct set of clinical outcomes. They also demonstrated that mosaic embryos' reproductive potential is affected by the complexity and the number of euploid cells in the TE biopsy. Compared to aneuploid mosaic embryos with one or two affected chromosomes, the embryos with a mosaic segmental aneuploidy had the best outcomes (implantation p<0.0001, OPR/BR p<0,0001). However, the implantation and the OPR/BR were less favorable compared to the euploid control group (51,3% vs 61,1%, p=0,0004 and 42,6%vs 52,7%, p=0,0003 respectively). The group with complex mosaic aneuploidy had the least favorable outcomes. Given these results, it is possible to draw a better stratification of mosaic embryos.”

Lines 260 – 279, page 6

We reformulated the following paragraph:

“These findings were consistent with the results previously obtained from the transfer of 100 mosaic embryos carried out in a single-center study [35]. This prospective study concluded that embryos carrying a single segmental abnormality should be preferred, followed by those with less severe mosaicism (45% vs. 36,4% IR and 39,4% vs. 27,3% fetal heartbeat, respectively). Most recently, Viotti et al., 2021, [47] presented their analysis which was based on multicenter data of 1000 transferred mosaic embryo outcomes. Thus far, this is the largest dataset with mosaic embryo transfer outcomes. They confirmed that combined mosaic embryos (segmental and chromosomal abnormalities) have statistically significant lower implantation (45,5%) and higher OPR/BR (37%) than euploid embryos (57,2% and 52,3%, respectively). They also found that the level and type of mosaicism significantly affects the embryo transfer outcome. Mosaic embryos with segmental aneuploidy had significantly better clinical results, followed by the group with one affected chromosome and by the group with three or more affected chromosomes (complex group) (implantation 51,6% vs. 46,4 vs. 30,4% and OPR/BR 43,1% vs. 34,8 vs. 20,8%). Furthermore, no significant differences between mosaic monosomies and trisomies were observed regarding the ploidy status of the mosaic embryos. The authors concluded that mosaic embryos also develop into physiologically healthy babies and proposed a classification system for these embryos. Finally, this ranking system can be accessed as a freely available web-based tool (https://embryoscore.web.app).”

We also replaced the reference Viotti et al. 2020 at the ASRM meeting with the new published article Viotti et al. 2021

  1. Viotti, M.; Victor, A.R.; Barnes, F.L.; Zouves, C.G.; Besser, A.G.; Grifo, J.A.; Cheng, E.-H.; Lee, M.-S.; Horcajadas, J.A.; Corti, L.; et al. Using Outcome Data from One Thousand Mosaic Embryo Transfers to Formulate an Embryo Ranking System for Clinical Use. Fertility and Sterility 2021, 115, 1212–1224, doi:10.1016/j.fertnstert.2020.11.041.

Point 4

It should be mentioned that mosaicism could be normal and physiological phenomena in human embryos and probably all or majority of embryos are mosaic ?? Do we PGTA laboratories overdiagnose or misdiagnose  something?  Do  practitioners  “lost babies ” discarding  an embryos from FET ? It would be interesting  to see  more discussion and authors opinion in this review.  

 Response 4

We very much appreciate the encouraging suggestions and agree that these comments would improve the review as they add a more critical perspective.

The following paragraphs have been added in the document to reflect these points:

Lines 218-227, page 5

“Based on mouse model findings [39], it has been shown that aneuploid embryos may self-correct downstream by extruding abnormal blastomeres as cell debris [43]. Recently, in another research setting, where single-cell genomic data (scRNA-seq) was used to quantify mosaicism in human preimplantation-stage embryos, it was concluded that low-level mosaicism is a frequent phenomenon, whereas high-level mosaicism is relatively uncommon [44]. Single-cell analysis of human embryos reveals diverse patterns of aneuploidy and mosaicism. These observations demonstrate that mosaicism in blastocyst-stage embryos is a common characteristic that can be found in almost all embryos. It seems that early-stage embryos are dynamic systems with the ability to self-correct.”

Lines 347-389, pages 8-9

“While new technological innovations brought important improvements in reliably detecting various types of genetic errors, such as mosaicism, the interpretation of PGT-A results faces new challenges. The selection of embryos for transfer is more complex, and the need for defining more specific criteria for a clinical diagnosis is more evident. It is important to examine and address the limitations of this new technology and set new guidelines in applying uniform reporting practices. It is necessary to recognize that there is great variability regarding the definitions and transfer thresholds of mosaic embryos and that the biopsied samples may not always represent the chromosomal state of the entire embryo. Furthermore, damage or loss of blastocysts from the TE biopsy and errors occurring during the genetic analysis of the small amount of DNA may impact the reliability of mosaic diagnosis. Due to these technical and inherent limitations, some normal embryos with potential for normal euploid pregnancies, if transferred, are discarded after PGT-A. These false-positive results can cause a decrease in life-birth rates.

The introduction of NGS technology allows the identification and reporting of intermediate chromosomal copy numbers. Therefore, there is a trend towards changing terminology on the definition of mosaicism. The American Society for Reproductive Medicine (ASRM) committee uses the phrase "embryos with intermediate copy numbers" when they refer to embryos diagnosed as mosaics [54]. A published opinion by Paulson and Treff 2020, proposed that the latter term is more accurate [55]. The designation “mosaic” should be replaced since it is inaccurate and misleading about its clinical significance.

As mentioned earlier [48], more than 100 healthy life births after mosaic embryo transfer have been reported. Based on these observations, it can be concluded that embryonic trophectoderm mosaicism may represent a normal variant of early embryo development and not a pathological feature. On the contrary, one cannot oversee that low-level mosaic embryos appear to have better clinical outcomes than mosaic embryos with high-level mosaicism. Besides, embryos categorized as low-level mosaics have significantly poorer clinical outcomes than the euploid group.

It is important to point out that there are limited data regarding neonatal and postnatal outcomes of mosaic embryo transfers. The results as regards IR, OPR, and LBR are heterogeneous. This highlights the need for consistent follow-up data after transfer-ring mosaic embryos with clinical and genetic outcomes. The conduction of larger-scale multicenter follow-up studies will contribute to the risk assessment of mosaic embryo transfer and help find a balance between increasing the rates of favorable clinical outcomes and decreasing the exclusion of embryos with implantation potential.                          

However, a recently updated Practice Committee Opinion of the ASRM and the Society’s Genetic Counseling Professional Group (GCPG) does ‘not endorse, nor does it suggest that PGT-A is appropriate in all cases of in vitro fertilization’ [54]. The importance of patient counseling must be emphasized, and flexibility and individualized treatment strategies should be strongly considered in clinical practice combined with a poly-parametric approach to reach decision making.”  

In addition, we have added the respective references:

  1. Orvieto, R.; Shimon, C.; Rienstein, S.; Jonish-Grossman, A.; Shani, H.; Aizer, A. Do Human Embryos Have the Ability of Self-Correction? Reproductive Biology and Endocrinology 2020, 18, 98, doi:10.1186/s12958-020-00650-8

  1. Starostik, M.R.; Sosina, O.A.; McCoy, R.C. Single-Cell Analysis of Human Embryos Reveals Diverse Patterns of Aneuploidy and Mosaicism. Genome Res. 2020, doi:10.1101/gr.262774.120.

  1. Abhari, S.; Kawwass, J.F. Pregnancy and Neonatal Outcomes after Transfer of Mosaic Embryos: A Review. J Clin Med 2021, 10, doi:10.3390/jcm10071369.

  1. Practice Committee and Genetic Counseling Professional Group (GCPG) of the American Society for Reproductive Medicine. Electronic address: asrm@asrm.org Clinical Management of Mosaic Results from Preimplantation Genetic Testing for Aneuploidy (PGT-A) of Blastocysts: A Committee Opinion. Fertil Steril 2020, 114, 246–254, doi:10.1016/j.fertnstert.2020.05.014.

  1. Paulson, R.; Treff, N. Isn’t It Time to Stop Calling Preimplantation Embryos “Mosaic”? F&S Reports 2020, 1, 164–165, doi:10.1016/j.xfre.2020.10.009

Reviewer 2 Report

Early human embryos naturally conceived or assisted reproductive technologies (ART) often display a high incidence of chromosomal abnormalities. Embryos viability might be compromised by chromosomal copy number deviations or chromosome structural rearrangements.  Technical improvements permit the identification of chromosomal mosaicism lately. Classically, embryos identified with chromosomal abnormalities before intrauterine transfer were excluded, thus lowering the available ones to be transferred.

The topic of the clinical transfer of embryos classified as a mosaic by PGT-A was first issued in 2015 in cases with no available euploid embryos. Since then, more data sustain this approach, as healthy babies might result from such embryos.

The current article is well written and approaches the problem of mosaic embryos that are reported to result in lower rates of implantation and a higher likelihood of miscarriage than euploid embryos but may lead to births with no overt medical conditions.

Point 1. Maybe it would be beneficial to analyze this subject also from the perspective of obstetrical and neonatal outcomes, as recently are available reports on the matter.

  1. Abhari S, Kawwass JF. Pregnancy and Neonatal Outcomes after Transfer of Mosaic Embryos: A Review. J Clin Med. 2021;10(7):1369. Published 2021 Mar 27. doi:10.3390/jcm10071369
  2. Ying Xin Zhang 1,† , Jang Jih Chen 2,† , Sunanta Nabu 3,† , Queenie Sum Yee Yeung 1 , Ying Li 1 , Jia Hui Tan 2 , Wanwisa Suksalak 3 , Sujin Chanchamroen 3 , Wiwat Quangkananurug 3 , Pak Seng Wong 2 , Jacqueline Pui Wah Chung 1 and Kwong Wai Choy. The Pregnancy Outcome of Mosaic Embryo Transfer: A Prospective Multicenter Study and Meta-Analysis. Genes 2020, 11, 973; doi:10.3390/genes11090973

Author Response

Response to Reviewer 2 Comments

Early human embryos naturally conceived or assisted reproductive technologies (ART) often display a high incidence of chromosomal abnormalities. Embryos viability might be compromised by chromosomal copy number deviations or chromosome structural rearrangements.  Technical improvements permit the identification of chromosomal mosaicism lately. Classically, embryos identified with chromosomal abnormalities before intrauterine transfer were excluded, thus lowering the available ones to be transferred.

The topic of the clinical transfer of embryos classified as a mosaic by PGT-A was first issued in 2015 in cases with no available euploid embryos. Since then, more data sustain this approach, as healthy babies might result from such embryos.

The current article is well written and approaches the problem of mosaic embryos that are reported to result in lower rates of implantation and a higher likelihood of miscarriage than euploid embryos but may lead to births with no overt medical conditions.

Point 1. Maybe it would be beneficial to analyze this subject also from the perspective of       obstetrical and neonatal outcomes, as recently are available reports on the matter.

  1. Abhari S, Kawwass JF. Pregnancy and Neonatal Outcomes after Transfer of Mosaic Embryos: A Review. J Clin Med. 2021;10(7):1369. Published 2021 Mar 27. doi:10.3390/jcm10071369
  2. Ying Xin Zhang 1,† , Jang Jih Chen 2,† , Sunanta Nabu 3,† , Queenie Sum Yee Yeung 1 , Ying Li 1 , Jia Hui Tan 2 , Wanwisa Suksalak 3 , Sujin Chanchamroen 3 , Wiwat Quangkananurug 3 , Pak Seng Wong 2 , Jacqueline Pui Wah Chung 1 and Kwong Wai Choy. The Pregnancy Outcome of Mosaic Embryo Transfer: A Prospective Multicenter Study and Meta-Analysis. Genes 2020, 11, 973; doi:10.3390/genes1109097

Response

We fully agree with the reviewers’ suggestion and we wish to thank the reviewer for bringing these articles to our attention. We would like to point out that the literature regarding the obstetrical and neonatal outcomes is very limited. The reviewer correctly highlighted these articles.

We have incorporated the reviewers’ suggestion and discussed these two references as follows:

Lines 173-176, page 4

“Similarly, in a recent multicenter prospective study, the authors demonstrated that mosaic embryo transfers compared to euploid embryos and non-PGT transfers revealed a significantly lower clinical PR (40.1% vs. 59.0% vs. 48.4%) and higher MR (33.3% vs. 20.5% vs. 27.4%) respectively [38].”

Lines 280-285, page 6

“According to the recently published review [48], more than 100 live births have been reported after mosaic embryo transfer. There were no detected differences regarding the birth weight, preterm delivery rate, or risk of congenital malformations in the examined newborns. However, much more data concerning perinatal and long-term neonatal outcomes born from mosaic embryos are imperative to draw definite conclusions and to provide optimal clinical guidance.”

We also included the following two references:

  1. Zhang, Y.X.; Chen, J.J.; Nabu, S.; Yeung, Q.S.Y.; Li, Y.; Tan, J.H.; Suksalak, W.; Chanchamroen, S.; Quangkananurug, W.; Wong, P.S.; et al. The Pregnancy Outcome of Mosaic Embryo Transfer: A Prospective Multicenter Study and Meta-Analysis. Genes (Basel) 2020, 11, doi:10.3390/genes11090973.

48. Abhari, S.; Kawwass, J.F. Pregnancy and Neonatal Outcomes after Transfer of Mosaic Embryos: A Review. J Clin Med 2021, 10, doi:1
